# Context Matters: Understanding Practitioner Priorities for Alignment in Large Language Models for Code

## Abstract

Large Language Models (LLMs) are increasingly used for code generation tasks. The LLM generated code may be functionally correct but could remain insecure, non-compliant, or non-maintainable. The alignment of code LLM is promising to ensure that the generated code satisfies both functional and non-functional requirements. However, we are aware of very little research on the software practitioners' (ML/software engineers) needs of LLM alignment for code generation. We report a survey of 30 practitioners from diverse industry and research backgrounds. We investigate how they prioritize functional versus non-functional code properties and what factors drive their alignment decisions. Around 75.9% of practitioners view non-functional properties as equally or contextually important alongside functional correctness. Task requirements (80.0%) and compute resources (73.3%) also influence alignment decision-making. We find that all the respondents considered the need of systematic guidance (e.g., domain-specific context) is necessary during alignment due it being a more niche area than fine-tuning.

## CCS Concepts

• **Computing methodologies** → **Artificial intelligence**; **Machine learning**; • **Software and its engineering** → **Software creation and management**; • **Security and privacy** → *Software and application security*.

## Keywords

LLM Alignment, Functional Requirements, Non functional Requirements, LLM Code Generation

**ACM Reference Format:**
Anonymous Author(s). 2018. Context Matters: Understanding Practitioner Priorities for Alignment in Large Language Models for Code . In *Proceedings of ACM International Conference on AI-powered Software (AIware)*. ACM, New York, NY, USA, 9 pages. https://doi.org/XXXXXXX.XXXXXXX

## 1 Introduction

Large Language Models (LLMs) have transformed the way developers build software [4, 34]. Tools such as ChatGPT [6], GitHub Copilot [5], and Claude [2] are now deeply embedded in the development process, assisting developers in writing, reviewing, and refactoring code [18, 34]. As adoption grows, LLM-generated code is becoming a significant portion of many codebases [4]. However,

this shift introduces new challenges: generated code may contain security vulnerabilities, exhibit poor readability, or violate established coding standards [24]. A function may compile and pass all test cases by satisfying functional correctness, and yet it may still introduce serious security and compliance risks in production.

```
Task: Retrieve a user record from the database

# Write a Python function that takes a username
# and returns the user from the database
def get_user(username):
```

```
Misaligned (DeepSeek Coder 1.3B Base): Vulnerable to SQL injection

def get_user(username):
    query = "SELECT * FROM users WHERE name = '" \
            + username + "'"
    return db.execute(query)
```

```
Aligned (DeepSeek Coder 1.3B Instruct): Parameterized query

def get_user(username):
    query = "SELECT * FROM users WHERE name = %s"
    return db.execute(query, (username,))
```

**Figure 1: Security alignment example: Given the same task prompt, the unaligned base model (DeepSeek Coder 1.3B Base) generates code vulnerable to SQL injection via string concatenation, while the instruction-tuned variant produces a parameterized query that prevents injection.**

To illustrate how these risks manifest, we prompted the unaligned DeepSeek Coder 1.3B Base model [16] to generate a function that retrieves a user record from a database. As shown in Figure 1, the base model directly concatenates user input into the SQL query. This function works correctly for normal inputs, but an attacker could supply a crafted input such as ' OR 1=1 – to access all user records, or worse, modify or delete data. In contrast, the instruction-tuned variant (DeepSeek Coder 1.3B Instruct) generates a parameterized query that separates user data from the query structure. Both functions produce identical results for legitimate inputs, but only the aligned version prevents SQL injection (CWE-89). This is consistent with prior findings that LLMs frequently generate insecure code patterns [10, 24], and that unaligned base models are particularly prone to reproducing vulnerable patterns present in their training data [26].

Now consider a developer working on a healthcare application who asks an LLM to generate a function that logs patient access events. As shown in Figure 2, a misaligned response writes sensitive patient data, including the full medical record, to a plaintext file. Under regulations such as HIPAA [1], this would be a compliance violation. An aligned model could be taught instead to log only anonymized identifiers, use a structured logging framework suitable

```
Misaligned: Logs sensitive data

def log_access(patient_name, doctor, record):
    with open("access.log", "a") as f:
        f.write(f"{doctor} accessed "
                f"{patient_name}: {record}\n")
```

```
Aligned: Compliant audit logging

import logging, hashlib
audit_logger = logging.getLogger("hipaa_audit")

def log_access(patient_id, doctor_id, record_id):
    audit_logger.info(
        "ACCESS patient=%s by=%s record=%s",
        hashlib.sha256(
            patient_id.encode()).hexdigest()[:8],
        doctor_id, record_id)
```

**Figure 2: Compliance alignment: Both functions log patient access, but only the aligned version uses anonymized identifiers and structured logging for regulatory audit trails.**

for audit trails, and avoid exposing sensitive information, all of which are expected practices in regulated environments [3].

These examples highlight a fundamental gap: functional correctness alone is not sufficient for production-quality code. Non-functional properties such as security and compliance are equally critical [7]. This is where LLM alignment could be used.

LLM alignment refers to techniques that train LLMs to prefer outputs matching specified quality criteria [23, 25, 27]. Originally developed to steer general-purpose LLMs away from harmful responses and toward human values [23], alignment relies on preference data: pairs of responses where a *chosen* response reflects desired behavior and a *rejected* response represents behavior to avoid. Common alignment methods include reward-based approaches such as Reinforcement Learning from Human Feedback (RLHF) [23] and reward-free approaches such as Direct Preference Optimization (DPO) [25]. When applied to code generation, the chosen response contains secure, readable, and standards-compliant code, while the rejected response contains code with vulnerabilities or quality issues.

Despite growing technical research on alignment methods for code [15, 23, 25, 30], a critical perspective remains missing: *what do software engineering practitioners actually need from code LLM alignment?* Existing studies focus on algorithmic improvements and benchmark scores, but do not investigate practitioner priorities, decision-making processes, or guidance needs. This gap matters because, as the examples above show, alignment is not a one-size-fits-all process. Different projects, domains, and development stages may require different trade-offs between functional correctness and non-functional properties like security or regulatory compliance. Prior survey-based work has studied developer use of LLM code generation tools [19] and programmer expectations [29], but no study has specifically investigated how practitioners think about alignment trade-offs or whether they want systematic guidance.

We address this gap with a practitioner-centered study. We surveyed 30 ML and software engineers from diverse industry organizations (e.g., H2O.ai, Synopsys, CIBC, WSO2) and academic institutions (e.g., Mila AI Institute). We investigated three research questions: (1) how practitioners prioritize functional correctness versus non-functional code properties, (2) what factors drive their alignment decision-making, and (3) whether they need for systematic alignment guidance. Our contributions are as follows:

(1) Our **Empirical evidence** shows that practitioners consider both functional and non-functional code properties important for alignment, with priorities varying by domain.
(2) We offer **a characterization** of how practitioners navigate alignment decisions, by identifying target task requirements and compute resources as the dominant decision factors.
(3) We present **quantitative evidences** of a strong and unmet demand for systematic alignment guidance, where 100% of respondents (excluding N/A) rating such guidance as useful.

## 2 Background & Related Work

**LLM Alignment.** refers to a technique that steer language models toward human preferences, originally introduced to align models with human values and away from harmful responses [27]. These techniques rely on preference data, which consists of two response columns: a chosen column containing expected responses that reflect aligned values, and a rejected column containing responses the model should learn to avoid, typically harmful or undesirable outputs [27]. Alignment techniques can be broadly categorized into two types: reward-based methods such as Reinforcement Learning from Human Feedback (RLHF) [23], and reward-free methods such as Direct Preference Optimization (DPO) [25]. While alignment was originally developed for general NLP tasks [27], it can also be applied to specific domains such as code generation [31]. In this setting, the preference data contains functionally correct code in the chosen column and code with issues in the rejected column.

**Starting Pathways.** When aligning an LLM for code-related tasks, practitioners must choose one of two starting pathways: Pre-Trained to Aligned (PTA) and Fine-Tuned to Aligned (FTA) . In the FTA pathway, an already instruction-tuned model is selected and further alignment is performed as needed. In the PTA pathway, supervised fine-tuning must first be applied before alignment can proceed. The choice between these pathways is determined by the practitioner depending on their specific use case.

**Code Requirements.** In code LLM alignment, the target properties in the model is aligned can be two types [7, 31]. The first is functional alignment, which targets generating functionally correct code for a given task. The second is non-functional alignment, which can target properties such as instruction following, coding style, and readability, as defined in [31].

**Code Generation Quality.** LLM-based code generation has been extensively studied for functional requirements, with benchmarks such as HumanEval [12], MBPP [8], EvoEval [32], and EvalPerf [20] primarily focusing on whether the generated code is functionally correct — that is, whether it compiles and produces correct outputs. However, relatively less research has focused on non-functional requirements. Recent work such as CodeUltraFeedback [31] explores this gap by defining five dimensions of code quality: instruction following, code readability, complexity/efficiency, coding style, and code explanation. Studies have also been conducted on secure versus vulnerable code generation [13, 17, 28, 33]. Despite these efforts,

non-functional [31] aspects remain underexplored, and the existing work tends to be purely technical without investigating what practitioners need or providing actionable recommendations.

**Practitioner Studies in SE.** Survey-based studies have been valuable for understanding software engineering practitioner needs. Vaithilingam et al. [29] studied programmer expectations of code generation, revealing gaps between what programmers want and what tools provide. Barke et al. [9] investigated how programmers interact with AI code assistants, identifying distinct usage patterns. However, no prior study has specifically investigated practitioner needs for alignment of code LLMs — that is, how practitioners think about the trade-offs between functional and non-functional properties, what factors drive their alignment decisions, and whether they want systematic guidance. Our work fills this gap.

## 3 Methodology

We designed and conducted a survey to understand practitioner perspectives on code LLM alignment. This section describes the survey methodology, collection approach, and analysis methods.

### 3.1 Research Questions

We answer three research questions, each targeting a distinct aspect of code LLM alignment from the practitioner perspective.

**RQ1.** *How do SE practitioners prioritize functional correctness versus non-functional code properties in LLM-generated code?* This question investigates what alignment should optimize for, from the practitioner perspective. We examine both explicit priority ratings and choices in concrete scenarios.

**RQ2.** *What factors drive practitioner decision-making when configuring LLM alignment for code generation tasks?* This question investigates *how* practitioners currently navigate alignment decisions; what starting points they prefer, what factors they consider, and how they evaluate outcomes.

**RQ3.** *To what extent do practitioners perceive a need for systematic guidance on code LLM alignment?* This question investigates *where the gaps are*; whether practitioners want structured recommendations and what form these should take.

### 3.2 Survey Design

Our survey consists of 14 questions organized into four sections, as shown in Table 1. The survey was designed to answer the three research questions defined above. Notably, Q5 presents a concrete scenario in which practitioners choose between *Option A* (40% relative improvement in non-functional properties, reaching 80% overall performance) and *Option B* (5% relative improvement, reaching 85% overall with standard code quality).

### 3.3 Participants

We used a snowball [14] approach to gather participants from diverse organizations and academic institutions. Participants were contacted through professional networks and research collaborations, targeting individuals with experience in software development, machine learning, or LLM-related work. Figure 3 summarizes participant demographics across three dimensions.

In terms of role distribution, participants include ML Engineers/Researchers (40.0%, n = 12), Software Engineers (33.3%, n = 10), and other roles including Research Scientists, Graduate Students, Engineering Managers, and DevOps practitioners (26.7%, n = 8). Regarding experience, the majority have 3–5 years (53.3%, n = 16), followed by 1–2 years (26.7%, n = 8), less than 1 year (13.3%, n = 4), and more than 5 years (6.7%, n = 2). With respect to alignment experience, participants span the full spectrum: 30.0% (n = 9) have no alignment experience, 23.3% (n = 7) are familiar with the process but have not done it, 23.3% (n = 7) have done it once or twice, and 23.3% (n = 7) have done it multiple times. This distribution ensures we capture perspectives from both alignment-experienced and alignment-naive practitioners. Participants come from diverse organizations including AI companies, financial services, technology firms, and academic institutions across multiple countries.

### 3.4 Analysis Methods

Our analysis combines quantitative and qualitative approaches. For closed-ended questions, we report descriptive statistics including frequencies and percentages. We report exact $p$-values and use $\alpha = 0.05$ as the significance threshold. Given the sample size ($N = 30$), we interpret non-significant results as evidence of *consistency* across groups that is, the absence of group differences is itself informative.

For open-ended questions (Q6, Q11, Q14), we employ thematic analysis following Braun and Clarke's six-phase approach [11, 22]: familiarization, initial coding, theme searching, theme reviewing, theme defining, and report production.

### 3.5 Threats to Validity

**Internal Validity.** Our findings rely on self-reported data, which may not fully reflect actual behavior. To mitigate this, we included open-ended questions (Q6, Q14) that provide qualitative context validating the quantitative patterns, and our sample includes 46.7% of practitioners with hands-on alignment experience, grounding responses in real practice rather than speculation.

**External Validity.** Our sample size (N=30) limits generalizability. However, we deliberately recruited across diverse roles (ML engineers, software engineers, research scientists), experience levels, and organization types (industry and academia) using snowball sampling to maximize coverage. Our findings also align with intuitions from related work on developer decision-making in software engineering, suggesting broader applicability.

**Construct Validity.** Terms like "non-functional properties" may be interpreted differently across respondents. We mitigated this by providing concrete examples (security, readability, compliance) in survey questions and by including scenario-based questions (Q5, Q6) that ground abstract constructs in specific decision contexts.

## 4 Results

### 4.1 RQ1: Functional vs. Non-Functional Priorities during Code LLM Alignment

**Explicit Priority Ratings (Q4).** When asked directly how they prioritize functional correctness versus non-functional properties, practitioners showed nuanced views rather than a binary preference. The largest groups indicated that priorities are context-dependent: 37.9% (n = 11) said "it varies by project context" and 37.9% (n = 11) said "both are equally important." Only 13.8% (n = 4) prioritized functional correctness and 10.3% (n = 3) prioritized non-functional

**Table 1: Survey questions organized by research question theme. MC = multiple choice, MS = multi-select, OE = open-ended.**

| ID | Question | Answer Options | Type |
|---|---|---|---|
| *Demographics* | | | |
| Q1 | What is your primary role? | ML Engineer/Researcher; Software Engineer; Research Scientist; Engineering Manager/Tech Lead; Student; Other | MC |
| Q2 | How many years of experience do you have in the software industry? | Less than 1 year; 1–2 years; 3–5 years; More than 5 years | MC |
| Q3 | Have you performed fine-tuning or alignment training on LLMs? | No experience; Done once or twice; Familiar but not done; Done multiple times | MC |
| *RQ1: Functional vs. Non-Functional Priorities* | | | |
| Q4 | How do you prioritize functional correctness vs. non-functional properties (security, instruction-following, compliance)? | Functional correctness more important; Non-functional more important; Both equally important; Varies by project context; Not applicable | MC |
| Q5 | Consider two alignment outcomes: (A) 40% relative improvement in non-functional properties, 80% overall; (B) 5% relative improvement, 85% overall. Which do you prefer for production? | Option A (larger non-functional improvement, lower overall); Option B (smaller improvement, higher overall); Cannot decide without more context | MC |
| Q6 | Describe a scenario where you would prioritize non-functional properties over functional correctness, or vice versa. | *Free-text response* | OE |
| *RQ2: Alignment Decision-Making Factors* | | | |
| Q7 | When adapting an LLM for code-related tasks, which starting point do you or your team typically use? | Base pretrained model (PTA); Instruction-tuned model (FTA); Evaluate both depending on use case; Not applicable | MC |
| Q8 | How important do you consider the choice between starting from a pretrained model (PTA) vs. an instruction-tuned model (FTA) for alignment outcomes? | Very important; Somewhat important; Not very important; Not applicable | MC |
| Q9 | What factors influence your choice of starting point? (Select all that apply) | Target task requirements; Compute resources; Time constraints; Expected improvement magnitude; Risk of degradation; Prior experience; Not considered; Standard practice | MS |
| Q10 | When evaluating alignment success, which metric do you prioritize more? | Absolute final performance; Relative improvement over baseline; Both, prioritize absolute; Both, prioritize relative; Not applicable | MC |
| Q11 | Have you encountered a situation where the choice of starting point (pretrained vs. instruction-tuned) affected your results? If yes, please briefly describe. | *Free-text response* | OE |
| *RQ3: Guidance Needs* | | | |
| Q12 | Would systematic recommendations for selecting alignment pathways (pretrained vs. instruction-tuned) be useful? | Very useful; Somewhat useful; Not applicable | MC |
| Q13 | Would systematic recommendations for balancing functional vs. non-functional requirements during alignment be useful? | Very useful; Somewhat useful; Not applicable | MC |
| Q14 | Any additional comments on alignment practices or trade-offs you have observed? | *Free-text response* | OE |

properties (excluding 1 N/A response, effective $N = 29$). This means that 75.9% of practitioners do not dismiss non-functional properties as secondary to functional correctness. Instead, the majority treat them as equally important or weigh them contextually.

**Concrete Scenario Choice (Q5).** To test whether these stated beliefs translate into decision-making, we presented participants with a concrete trade-off scenario. The results revealed a split: 46.7% ($n = 14$) chose Option A (larger non-functional improvement, lower absolute score) while 36.7% ($n = 11$) chose Option B (smaller improvement, higher absolute score). Five respondents (16.7%) could not decide without more context. Among those who expressed a preference ($N = 25$), 56.0% favored Option A. A binomial test

found this preference not statistically significant ($p = 0.690$), indicating no overwhelming consensus toward either option. This division reinforces rather than contradicts the explicit priority ratings: practitioners equally weigh both dimensions, and reasonable professionals can reach different conclusions depending on what they value.

**Qualitative: Domain-Driven Priorities (Q6).** Given this lack of consensus, what factors drive individual prioritization decisions? Open-ended responses to Q6 ("Describe a scenario where you would prioritize non-functional properties over functional correctness, or vice versa") provided insight. Of 30 respondents, 17 provided scenario descriptions (56.7% response rate). Thematic coding revealed

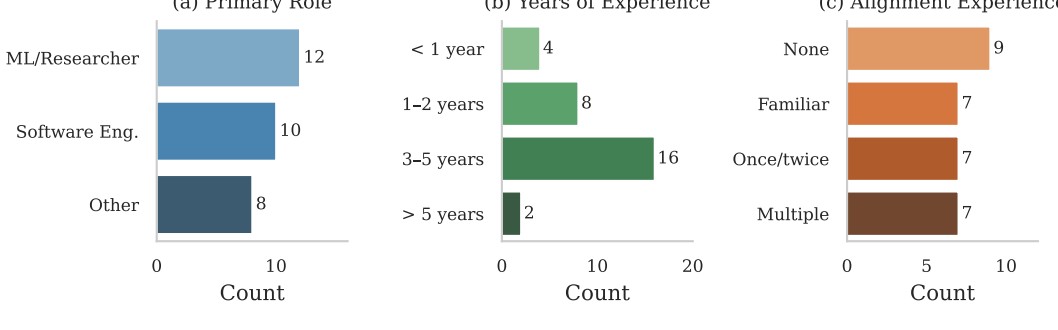

Figure 3: Participant demographics (N=30). (a) Roles grouped as ML Engineer/ML Researcher (12), Software Engineer (10), and Other including Research Scientist, Graduate Student, Engineering Manager, DevOps (8). (b) Years of industry experience. (c) Experience with LLM alignment training.

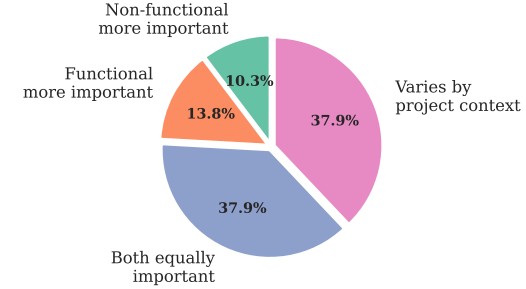

Figure 4: RQ1: Practitioner priorities. Most practitioners view the balance as context-dependent or equally important.

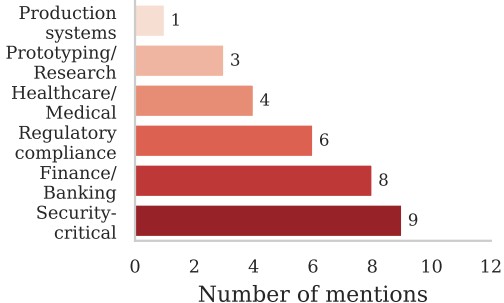

Figure 5: Domain themes from Q6 scenario descriptions (N=17 respondents). Security-critical and finance/banking contexts dominate, consistently favoring non-functional properties.

that domain context is the primary driver of prioritization, with three distinct themes emerging as shown Figure 5

*Theme 1: Regulated domains demand non-functional priority.* The most frequent theme (15/17 responses) linked non-functional prioritization to regulated or sensitive domains. Security-critical systems were mentioned by 9 respondents, finance/banking by 8, regulatory compliance by 6, and healthcare by 4. Representative quotes include:

> "In a banking or healthcare application, I would prioritize security and readability over functional correctness."

> "In a trading platform we prioritize security, readability, and compliance before release, even if the feature already works, because unsafe or illegal code can cause big losses."

*Theme 2: Development lifecycle stage shifts priorities.* Three respondents described a temporal pattern: functional correctness takes priority during prototyping and early development, while non-functional properties become critical for production:

> "For quick exploratory data analysis or proof-of-concept scripts, I prioritize functional correctness […] However, for production code or systems handling sensitive data […], I prioritize non-functional properties like security and readability."

*Theme 3: Non-functional as non-negotiable in user-facing systems.* Several respondents framed specific non-functional properties (especially security) as absolute requirements rather than trade-offs:

> "It is more important for the code for a social media app to secure user data (e.g., protect against SQL injection), than to function correctly."

**RQ1 Summary**

Practitioners overwhelmingly (75.9%) view non-functional properties as equally or contextually important alongside functional correctness. When forced to choose in concrete scenarios, a significant amount (56.0% of decided respondents) prefer larger non-functional gains over higher absolute performance; though the split reflects professional disagreement rather than methodological noise. The qualitative data explains this variation: domain context; especially security, finance, and healthcare; is the primary driver of prioritization, with lifecycle stage and user-facing requirements serving as secondary factors.

## 4.2 RQ2: Decision-Making Factors

In practice, practitioners must make concrete decisions about how to configure and execute alignment processes that directly shape whether those desired outcomes can be achieved. RQ2 investigates the practical side of alignment: what starting points practitioners

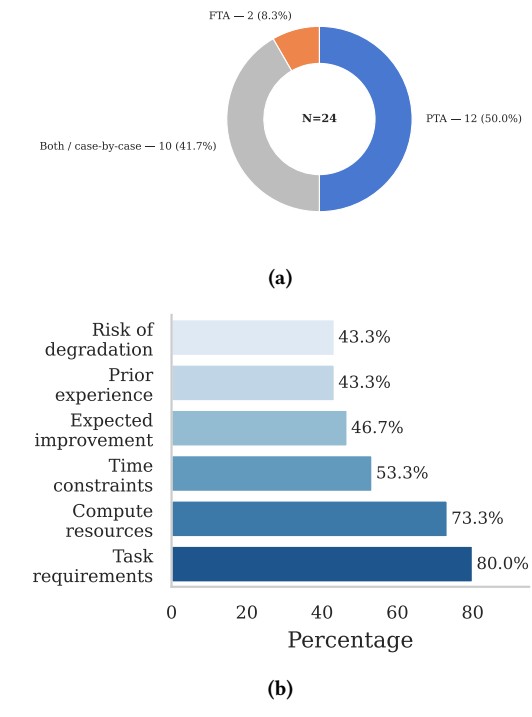

**(a)**

**(b)**

**Figure 6: RQ2: Alignment decision factors. (a) Starting point preferences among applicable respondents. (b) Factors ranked by selection frequency (multi-select Q9).**

choose, what factors influence these choices, how they evaluate success, and what challenges they encounter in real-world settings.

This section examines five key aspects of alignment decision-making: starting point preferences between pretrained and instruction-tuned models (Q7), the perceived importance of this choice (Q8), the specific factors that drive pathway selection (Q9), evaluation metric preferences (Q10), and real-world experiences that illuminate these decisions in practice (Q11).

*4.2.1 Starting Point Preferences (Q7).* practitioners favor base pretrained models (PTA pathway, 40.0%, $n = 12$), followed by evaluating both options depending on the use case (33.3%, $n = 10$). Only 6.7% ($n = 2$) consistently start with instruction-tuned models (FTA). The remaining 20.0% ($n = 6$) indicated the question was not applicable to their work.

Among those with an applicable starting point preference ($N = 24$) as shown in Figure 6a the PTA pathway is the most common choice (50.0%), but one-third explicitly adopt a case-by-case evaluation strategy. This suggests that while PTA is a common default, practitioners recognize the value of adaptive decision-making.

*4.2.2 Importance of Pathway Choice (Q8).* Given this preference for PTA pathways, we next examined how critical practitioners perceive this choice to be. An overwhelming 86.7% of respondents rated the choice between PTA and FTA pathways as "Very important" (40.0%, $n = 12$) or "Somewhat important" (46.7%, $n = 14$). Only one respondent (3.3%) rated it "Not very important."

*4.2.3 Decision Factors (Q9).* Having established that pathway choice matters for practitioners, we investigated what specific factors drive these decisions. Figure 6b ranks the factors practitioners consider when choosing an alignment pathway. Target task requirements lead (80.0%, $n = 24$), followed closely by available compute resources (73.3%, $n = 22$). Time constraints (53.3%, $n = 16$), expected improvement magnitude (46.7%, $n = 14$), prior experience with models (43.3%, $n = 13$), and risk of performance degradation (43.3%, $n = 13$) follow.

The dominance of task requirements and compute resources is notable: these are practical, project-specific factors rather than theoretical considerations. The lower ranking of risk of performance degradation (43.3%) is another notable pattern, given that degradation is a well-documented concern in alignment research [21].

*4.2.4 Evaluation Metric Preferences (Q10).* We examined how practitioners assess whether alignment was successful. Among respondents for whom the question was applicable ($N = 27$), practitioners were nearly evenly split between relative-leaning (55.6%, $n = 15$) and absolute-leaning (44.4%, $n = 12$) evaluation metrics. This confirms that both relative improvement and absolute performance are valued across roles, and evaluation frameworks should report both.

*4.2.5 Qualitative: Real-World Experiences (Q11).* To contextualize these quantitative patterns, we analyzed open-ended responses describing practitioners' real-world experiences with pathway decisions. Seven respondents (23.3%) shared real-world experiences with PTA versus FTA choices. Key patterns include:

**Pattern 1: Domain-specific tasks favor PTA.** Three respondents found pretrained models more effective for controlled, domain-specific tasks:

> "Pretrained models worked better for controlled, domain-specific extraction tasks, while instruction-tuned models performed better for interactive or conversational use cases."

**Pattern 2: Instruction-tuned models provide better starting structure.** Five respondents noted instruction-tuned models produce more reliable, well-structured code:

> "Instruction-tuned models provide more reliable, well-structured code with better documentation from the start."

**Pattern 3: Security-critical applications require careful pathway selection.** One detailed response described security outcomes varying by pathway:

> "Starting with an instruction-tuned model provided higher baseline performance (85%) but limited improvement potential. [. . . ] Starting from a pretrained model allowed us to achieve a +50% relative improvement in security alignment."

These real-world accounts reinforce our quantitative findings: task requirements drive pathway selection (Pattern 1), and practitioners weigh both starting point advantages (Pattern 2) and improvement potential (Pattern 3) when making decisions.

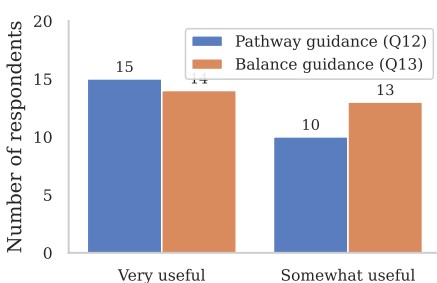

Figure 7: RQ3: Perceived usefulness of systematic alignment guidance (excluding N/A responses). Both types achieve 100% positive ratings among applicable respondents.

---

**RQ2 Summary**

Practitioners favor PTA pathways (50.0% of applicable respondents) but one-third evaluate both options per use case. Task requirements (80.0%) and compute resources (73.3%) are the dominant decision factors. The choice between relative and absolute metrics is evenly split, and these preferences are consistent across roles and experience levels. Real-world experiences reveal that domain-specific requirements and security considerations significantly influence pathway selection.

---

## 4.3 RQ3: Guidance Needs

Given the complexity and context-dependency of alignment decisions, do practitioners perceive a need for systematic guidance? RQ3 investigates whether there is demand for structured recommendations, who wants this guidance, and what form it should take.

This section examines three key aspects: the perceived usefulness of pathway selection guidance (Q12), the demand for recommendations on balancing functional versus non-functional requirements (Q13), whether guidance needs vary by experience level, and qualitative insights into what effective guidance should address (Q14).

*4.3.1 Demand for Pathway Recommendations (Q12).* We first examined whether practitioners would find systematic recommendations for choosing between PTA and FTA pathways useful. Among respondents for whom the question was applicable ($N = 25$), 100% found systematic recommendations for selecting alignment pathways useful: 60.0% ($n = 15$) rated them "Very useful" and 40.0% ($n = 10$) "Somewhat useful" (Figure 7).

*4.3.2 Demand for Balance Recommendations (Q13).* We investigated whether practitioners similarly desire help with the functional versus non-functional trade-offs identified in RQ1. Among applicable respondents ($N = 27$), 100% found recommendations for balancing functional versus non-functional requirements useful: 51.9% ($n = 14$) "Very useful" and 48.1% ($n = 13$) "Somewhat useful."

The universal demand for both types of guidance suggests that practitioners recognize alignment as a multifaceted challenge requiring support across multiple dimensions,not just pathway selection, but also priority balancing.

*4.3.3 Qualitative: What Guidance Should Address (Q14).* To understand what form effective guidance should take, we analyzed open-ended comments from five respondents who described specific needs. Responses are organized into three themes:

*Stability-plasticity awareness:*

"The stability-plasticity dilemma is crucial in determining pathway selection. [...] For production systems, the FTA pathway's stability often outweighs the PTA pathway's potential gains."

*Human-in-the-loop considerations:*

"Code generation does have a human in the loop. [...] So non-functional requirement alignment should be verified by the human developer. [...] Multi-step code generation with first generating functionally accurate code and then making it aligned with NFRs in the second pass are also viable options."

*Context-dependent trade-offs:*

"Alignment improves safety and usability but can reduce precision in specialized tasks, so it often needs to be balanced with task-specific constraints and validation."

These comments show that practitioners don't simply want prescriptive rules; they want frameworks that help them reason about stability-plasticity trade-offs, integrate human oversight, and adapt decisions to specific contexts.

---

**RQ3 Summary**

There is strong demand for systematic alignment guidance. 100% of applicable respondents find both pathway selection and balance recommendations useful ($p < 0.001$). This demand is consistent across experience levels ($p = 0.772$), indicating a broad need regardless of alignment expertise. Qualitative responses suggest guidance should address stability-plasticity trade-offs, human-in-the-loop workflows, and context-dependent decision-making.

---

## 5 Discussion

Our findings reveal that code LLM alignment is not a purely technical challenge but a sociotechnical one, shaped by domain context, organizational constraints, and practitioner practices. Figure 8 summarizes how our key findings map to stakeholder-specific recommendations. We discuss the recommendations below.

## 5.1 Implications for SE Practice

First, alignment is not one-size-fits-all. The 75.9% of practitioners who view functional and non-functional properties as equally important or context-dependent (RQ1) indicate that no single alignment configuration will satisfy all use cases. Code LLM alignment tools should therefore support configurable objectives that allow developers to specify whether they prioritize security, readability, or functional correctness for a given project.

This need for configurability extends to domain-specific considerations. Our qualitative findings reveal a clear pattern: practitioners in regulated domains (finance, healthcare, compliance) consistently prioritize non-functional properties, while those in prototyping or research contexts prioritize functional correctness. Alignment tools should support domain-aware settings for example, a "finance"

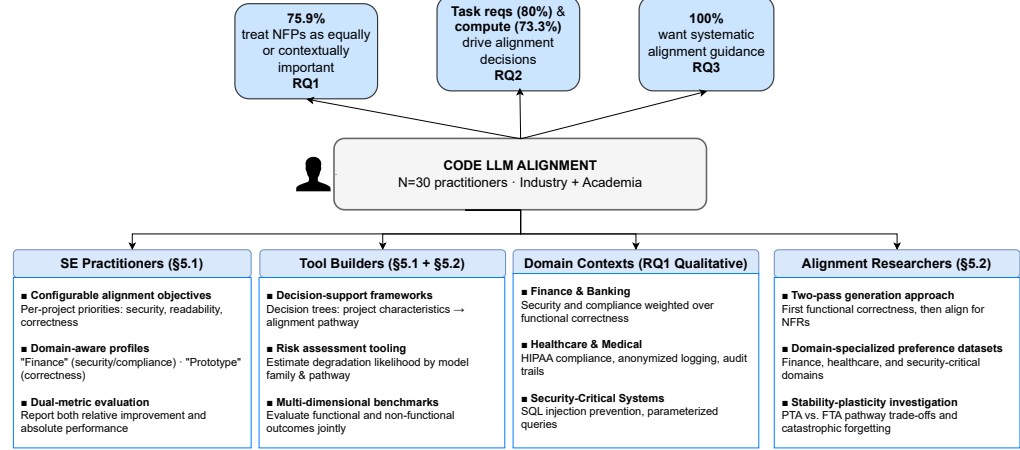

**Figure 8: Stakeholder implications mapping survey findings (RQ1–RQ3) to actionable recommendations for SE practitioners (5.1), alignment researchers (5.2), tool builders, and domain contexts.**

profile that weights security and compliance more heavily, or a "prototype" profile that focuses on functional correctness.

Finally, these diverse priorities also affect how alignment success should be measured. The near-even split between relative-leaning (55.6%) and absolute-leaning (44.4%) metric preferences (RQ2) validates the approach of reporting both metrics in alignment evaluations. Tool builders and researchers should present both relative improvement and absolute performance to support different practitioner decision-making styles.

## 5.2 Implications for Alignment Research

The demand for guidance (RQ3) represents a clear call to action. Currently, practitioners make alignment decisions based on task requirements and compute resources (RQ2), but lack frameworks to systematically evaluate trade-offs. We recommend developing decision trees mapping project characteristics to recommended alignment pathways, benchmark suites that evaluate outcomes across both functional and non-functional dimensions, and risk assessment tools that estimate degradation likelihood based on model family and pathway choice. Our qualitative data also suggests promising technical directions. One practitioner suggested a two-pass approach: first generate functionally correct code, then align for non-functional requirements. This idea deserves systematic investigation, as it could mitigate the trade-offs between functional and non-functional properties observed in empirical studies. More broadly, our findings highlight that finance, healthcare, and security domains have distinct alignment needs. Future work should investigate how alignment techniques perform in these specific contexts, potentially developing domain-specialized preference datasets.

## 5.3 Implications for Tool Builders

Our findings motivate three tool-level capabilities (Figure 8). The demand for guidance (RQ3) calls for *decision-support frameworks* such as decision trees mapping project characteristics to recommended alignment pathways. The low attention to degradation risk (43.3%, RQ2) despite its documented prevalence [21] suggests *risk*

*assessment tooling* that surfaces degradation likelihood based on model family and pathway choice. Finally, the near-even metric split (RQ2) and equal weighting of functional and non-functional properties (RQ1) require *multi-dimensional benchmarks* that jointly report correctness, security, readability, and compliance.

## 5.4 Implications for Specific Domains

RQ1 qualitative findings reveal domain-driven alignment needs (Figure 8). In *finance and banking* (8/17 respondents), practitioners consistently weight security and compliance over functional correctness. In *healthcare* (4/17), HIPAA compliance and anonymized logging are non-negotiable (Figure 2). In *security-critical systems* (9/17), vulnerabilities such as SQL injection (Figure 1) are treated as absolute requirements rather than trade-offs. These patterns reinforce the need for domain-aware profiles and specialized preference datasets recommended in Sections 5.1 and 5.2.

## 6 Conclusion

We present the first practitioner-centered study of code LLM alignment priorities and decision-making. Through a survey of 30 practitioners across diverse roles and organizations, we find that alignment cannot be reduced to a single optimization objective: 75.9% of practitioners treat non-functional properties as equally or contextually important alongside functional correctness. Task requirements and compute resources are the dominant factors guiding alignment pathway selection, and 100% of applicable respondents desire systematic guidance on both pathway selection and requirement balancing. These findings call for alignment tools with configurable, domain-aware objectives, evaluation frameworks with context-dependent trade-offs. Future work should develop such frameworks and investigate domain-specialized alignment approaches for security-critical, financial, and healthcare applications.

## 7 Data Availability

https://anonymous.4open.science/r/8805a959-9d92-4325-8b12-90c1580f4c75/

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
