# OpenReview forum: "Context Matters: Understanding Practitioner Priorities for Alignment in Large Language Models for Code"
_ACM.org/AIWare/2026/Conference — Submitted to AIware 2026_

### Official Review · Reviewer_3duF · 2026-03-08

**Rating:** 3
**Confidence:** 5

**Review:**

Strengths:

The study is relevant to the practical adoption of AI-generated code.

The study investigates a relevant problem in the field of Software Engineering.

Weaknesses:

The survey had a small number of participants.

The research could explore more details about projects when deciding between NFR and FR.

Comments:

The study is innovative and reports the relative importance of functional and non-functional requirements in projects, indicating that priority can vary depending on the project phase or domain. It can also reduce accuracy in specialized tasks (Santa Molison, Alfred, et al. "Is LLM-Generated Code More Maintainable & Reliable Than Human-Written Code?". 2025 ACM/IEEE International Symposium on Empirical Software Engineering and Measurement (ESEM). IEEE, 2025.)

I'm not sure why percentage values ​​were used in Q5.

The study could report whether there is a statistically significant difference in choice between PTA and FTA in RQ2.

Section 3.4 did not specify the statistical test used in the study.

When citing participants, please include the participant's anonymized ID in the respective citation.

Figure 8: Is the NFP in the upper left square an NFR?

The replication package contains the research results for future use in subsequent research.

**Summary:**

This article uses a survey of 30 professionals in the field to understand the alignment between functional requirements (FR) and non-functional requirements (NFR) for AI-generated code, exploring priority decisions and their alignment with quality and safety specifications. The results showed a balance between the relative importance of NFR and FR, but also that this importance depends on the specific tasks and complexity of the project.

---

> ### Author Response · Authors · 2026-03-20
>
> We thank the reviewer for their constructive feedback.
>
> **1. Small Number of Participants**
>
> We acknowledge that our sample size ($N=30$) is a limitation and have discussed this in Section 3.5 (Threats to Validity). However, we would like to highlight points that support the validity of our findings despite this sample size:
>
> *Diverse selection of survey responders.* We used snowball sampling to recruit across diverse roles (ML Engineers/Researchers, Software Engineers, Research Scientists, Engineering Managers), experience levels (ranging from $<$1 year to $>$5 years), and organization types (industry companies and academic institutions such as Mila AI Institute). This diversity helps mitigate the risk that findings reflect only a single organizational or role-based perspective.
>
> We agree this is a direction for future work. We will strengthen our discussion of this limitation in the revised manuscript.
>
> **2. Exploring More Details About Projects When Deciding Between NFR and FR**
>
> We agree that deeper exploration of project-level factors would strengthen the findings. Our current study captures domain-level context (e.g., finance, healthcare, security-critical systems) through Q6 and lifecycle-stage considerations through qualitative analysis (Section 4.1, Themes 1–3). However, we acknowledge that deeper project attributes such as project size, team composition, deployment environment, or regulatory policies were not systematically captured.
>
> In the revised manuscript, we will: (1) discuss these project-level factors as a concrete direction for future work, and (2) note how our stakeholder implications (Figure 8) can serve as a foundation for designing more granular studies that map specific project characteristics to alignment decisions.
>
> **3. Related Work on LLM-Generated Code Maintainability**
>
> We thank the reviewer for pointing us to this relevant work. The finding that alignment can reduce accuracy in specialized tasks is consistent with the trade-offs our practitioners describe (Section 4.1, Theme 2 and Section 4.3.3). We will incorporate this reference in the revised manuscript in Section 2 (Background \& Related Work) and in Section 5 (Discussion).
>
> **4. Rationale for Percentage Values in Q5**
>
> The percentages in Q5 were designed to create a concrete, quantified trade-off scenario accessible to practitioners across different experience levels. Option A represents significant non-functional improvement at the cost of lower absolute performance; Option B represents higher absolute performance with marginal non-functional gains. We will add a brief justification in the revised Section 3.2.
>
> **5. Statistical Test for PTA vs. FTA Choice in RQ2**
>
> In our current analysis, we report descriptive statistics for starting point preferences (Q7): PTA (50.0\%), Both/case-by-case (41.7\%), and FTA (8.3\%) among applicable respondents ($N=24$). We will add a chi-squared goodness-of-fit test to assess whether the distribution of preferences differs significantly from a uniform distribution. We will include this test result in the revised Section 4.2.1.
>
> **6. Specifying the Statistical Test in Section 3.4**
>
> In the current manuscript, we report exact $p$-values and use $\alpha = 0.05$ as the significance threshold (Section 3.4), and specific tests are mentioned where applied (e.g., binomial test in Section 4.1 for Q5). However, we agree that Section 3.4 could provide a more detailed description of all statistical tests used throughout the paper. In the revised manuscript, we will explicitly mention the tests applied.
>
> **7. Including Anonymized Participant IDs in Quotations**
>
> We agree that this is important for traceability. In the revised manuscript, we will assign each participant an anonymized identifier (e.g., P1–P30) and include the relevant ID alongside each direct quotation.
>
> **8. Typographical Error in Figure 8 (NFP vs. NFR)**
>
> Correct, this is a typographical inconsistency from our end. We will revise it in the revised manuscript.

---

### Official Review · Reviewer_Rcsj · 2026-03-08

**Rating:** 2
**Confidence:** 5

**Review:**

- The statement that “75.9% of practitioners view non-functional properties as equally or contextually important alongside functional correctness” seems consistent with long-standing software engineering practice. It is not clear what is new here specifically in the context of LLM-based code generation.
- As presented, the results almost suggest that developers are unsure what to prioritize, although this may be partly due to the limited context provided in the survey.
- The claim that “regulated domains demand non-functional priority” is generally expected in software engineering. Security, reliability, and compliance have always been critical in these domains, so it would be helpful to clarify what additional insight this study provides.
- For the quote “In a banking or healthcare application, I would prioritize security and readability over functional correctness,” it would be useful to know whether respondents actually work in these domains or are responding hypothetically. This is important to know whether these questions are representative of actual industrial practice?
- “ Code LLM alignment tools should therefore support configurable objectives that allow
developers to specify whether they prioritize security, readability, or functional correctness for a given project.” → The suggestion that alignment tools should allow developers to configure objectives such as security, readability, or correctness seems reasonable, but this also feels somewhat expected for most development tools. It would help if the paper clarified what this would look like in practice.
- “Alignment tools should support domain-aware settings for example, a “finance” → Similarly, the recommendation that alignment tools should support domain-aware settings (e.g., finance) is intuitive. The paper could benefit from explaining how such domain awareness would be implemented.
- The abbreviation FTA for instruction-tuned models is a bit confusing, since it does not obviously correspond to the term “instruction tuning.” Clarification would help.
- The statement that “code generation must have human in the loop” is already reflected in many current tools. For example, most modern AI coding assistants already rely on developer oversight.
- The reported context-dependent trade-offs may partly result from the limited context provided in the survey questions. In that sense, this appears to be more a limitation of the survey design than a limitation of LLMs themselves
- The observation that non-functional concerns often take precedence later in the software lifecycle is expected from a traditional software engineering perspective as well. Therefore, I wonder what additional insights this raises?
- The introduction spends a significant amount of space discussing correctness vs. security trade-offs. It is the normal that functional correctness != non functional quality. This section could likely be shortened.
- It would be helpful to clarify how the proposed notion of alignment differs from existing approaches such as RLHF, which already attempt to incorporate human preferences.
- The motivation for comparing pre-trained and instruction-tuned models is not entirely clear. In practice, most modern systems involve both pre-training and instruction tuning.
- For Q5, it is unclear how the specific trade-off percentages were chosen?
- In general, the survey questions lack motivations.
- Many of the conclusions (e.g., that trade-offs exist, non-functional requirements matter, alignment is important) reflect widely known principles in software engineering. It would help to highlight what the paper adds beyond these existing understandings.

**Summary:**

The paper investigates developers’ perspectives on trade-offs between functional correctness and non-functional properties in LLM-based code generation tasks, and explores how alignment tools may need to adapt to developer preferences. Through a survey of 30 practitioners, the study finds that developers often treat non-functional properties as equally important as functional correctness, with priorities varying depending on domain context, task requirements, and compute resources.

---

> ### Author Response · Authors · 2026-03-20
>
> We thank the reviewer for their constructive feedback.
>
> **Novelty of the 75.9% Finding**
>
> We acknowledge that the importance of nonfunctional requirements is well established in traditional SE. However, our contribution is its empirical validation in the context of LLM alignment for code generation. Our study provides evidence that practitioners expect alignment to address nonfunctional properties with comparable priority, highlighting a disconnect between what research optimizes for and what practitioners need.
>
> **Developer Uncertainty vs. Context-Dependence**
>
> We would like to clarify that the variation in responses reflects context dependence rather than confusion. The qualitative data from Q6 supports this: practitioners who said "it varies by project context" consistently provided specific, well reasoned scenarios (e.g., security in banking, correctness in prototyping). Those who selected "cannot decide without more context" in Q5 (16.7\%) were a minority.
>
> **Regulated Domains Demanding Non-Functional Priority**
>
> We agree that the principle is expected in traditional SE. The additional insight our study provides is that we quantify this preference specifically in the context of LLM alignment decisions, showing that practitioners in regulated domains expect alignment techniques to be configured differently.
>
> **Hypothetical vs. Actual Domain Experience of Respondents**
>
> Our participant pool includes practitioners from diverse organizations, including financial services and technology firms serving regulated industries. However, we did not systematically collect data on each respondent's specific domain. We acknowledge this as a limitation and will note it in the revised paper.
>
> **Configurable Alignment Objectives in Practice**
>
> One of the ways it could be implemented is to include weighted reward functions in RLHF where practitioners set relative weights for security, readability, and correctness, or profile-based DPO training with domain-curated preference datasets.
>
> **FTA Abbreviation Clarity**
>
> FTA stands for "Fine Tuned to Aligned," referring to the pathway where one starts from an already instruction tuned model and then applies alignment, in contrast to PTA ("Pre Trained to Aligned"). We agree the abbreviation could be clearer and will expand the definition.
>
> **Context-Dependent Trade-Offs Due to Survey**
>
> We agree this is a valid concern and have acknowledged it in Section 3.5 (Construct Validity). The limited context in survey questions may increase context-dependent responses. However, the open-ended responses (Q6, Q11, Q14) provide richer context and still show the same pattern.
>
> **Non-Functional Concerns in Later Software Lifecycle Stages**
>
> The insight we highlight is how it maps to LLM alignment strategy: alignment configurations could evolve with the project lifecycle, correctness-focused during prototyping, shifting to security- and compliance-weighted configurations before production. We will make this mapping more explicit in the revised paper.
>
> **Introduction Length**
>
> We agree. We will tighten the motivating examples and refocus the introduction on the research gap and contributions.
>
> **Distinction from RLHF and Existing Alignment Approaches**
>
> We want to clarify that our study does not propose a new alignment technique. Rather, we investigate how practitioners think about configuring and applying existing approaches (including RLHF and DPO) for code generation tasks. We will make this distinction clearer in the revised paper.
>
> **Motivation for Comparing Pre-Trained and Instruction-Tuned Models**
>
> The distinction matters because practitioners must choose a starting point for alignment, with practical implications for compute cost, improvement potential, and risk of catastrophic forgetting. Our data shows practitioners are split (50\% PTA, 41.7\% case by case) and consider it important (86.7\% rate it as very or somewhat important).
>
> **Q5 Trade-Off Percentages**
>
> The percentages in Q5 create a concrete, quantified tradeoff scenario accessible to practitioners across experience levels. Option A represents significant nonfunctional improvement at the cost of lower absolute performance; Option B represents higher absolute performance with marginal nonfunctional gains. We will add a brief justification in the revised Section 3.2.
>
> **Contributions Beyond Known SE Principles**
>
> Our contributions go beyond restating known SE principles: (1) we provide empirical evidence that these principles are actively expected by practitioners in the context of LLM alignment for code, where research has mainly focused on functional correctness, (2) we identify specific decision factors (task requirements at 80\%, compute resources at 73.3\%) actionable for the alignment community, (3) the demand for systematic guidance (100\% of applicable respondents) highlights a gap between research tooling and practitioner needs. We will adjust the framing to foreground these contributions more clearly.

---

### Official Review · Reviewer_TrBJ · 2026-03-11

**Rating:** 1
**Confidence:** 5

**Review:**

Strengths:
-------------------------
+ The topic is highly timely and relevant, addressing practical alignment of code LLMs rather than only benchmark performance or algorithmic advances.
+ Figure 8's mapping of findings to distinct stakeholder groups (SE practitioners, alignment researchers, tool builders, domain contexts) is a practical and well-structured contribution
+ The dataset is made available, and the paper is nicely written and easily understandable

Weaknesses:
--------------------------
- The central findings are plausible and useful, but not surprising. It is common and predictable that practitioners care about non-functional properties, priorities depend on the domain or context, compute and task needs affect model choices, and practitioners want guidance. No unexpected phenomenon is uncovered. The contribution is mainly descriptive.
- The paper states that it uses thematic analysis. However, considering the dataset size the claiming to thematic saturation is somewhat not convincing. Moreover, the paper also does not provide enough detail on: coding procedure, coder involvement, disagreement resolution, or how themes were derived and validated.
- The headline result that 100% of applicable respondents want systematic guidance is likely to attract attention. However, it excludes N/A responses, and comes from a small sample. It is unsurprising that practitioners would say guidance is useful. Also the options only contain  Useful and somewhat useful, " which seems the participants were biased to select useful options. In such cases, five scale likert-chart is best.
- The paper states that, given the sample size, non-significant results are interpreted as evidence of consistency across groups. That is methodologically weak. A non-significant result in a small-N study usually means insufficient evidence of a difference, not evidence of consistency. some conclusions sound stronger than the analysis can support.
- The study recruits participants through snowball sampling via “professional networks and research collaborations,” which creates a substantial risk of selection bias. This strategy is likely to over-represent individuals already connected to academic, research, or LLM-focused communities, and therefore more familiar with alignment concepts than the broader population of software practitioners. The participant pool reflects this concern: 40% of respondents are ML Engineers/Researchers and 33.3% are Software Engineers, suggesting a sample tilted toward technically specialized and research-adjacent perspectives rather than a representative cross-section of everyday practitioners using LLM-based coding tools. As a result, the external validity of the findings is limited, particularly where the discussion extrapolates to “SE practitioners” and “tool builders” in general. Although the paper briefly notes sampling limitations in the threats section, it does not sufficiently address how this recruitment strategy may have shaped the responses or narrowed the applicability of the conclusions.
- The trade-off scenario in Q5 is confounded because it varies both the magnitude of improvement in non-functional properties and the absolute overall performance level at the same time. Option A offers a 40% relative improvement in non-functional properties with 80% overall performance, whereas Option B offers only a 5% relative improvement but 85% overall performance. As a result, participant choices cannot be cleanly interpreted as preferences over non-functional properties alone. A respondent may prefer Option B not because they value non-functional gains less, but because 85% overall performance exceeds their acceptable threshold whereas 80% does not. The fact that 16.7% of respondents reported that they could not decide without more context directly reinforces this interpretation problem. Consequently, the reported result that “56% prefer larger non-functional gains” is difficult to interpret, since the responses may instead reflect minimum acceptable performance requirements or threshold effects rather than genuine prioritization of non-functional properties.
- Section 5 presents several design and research recommendations that are plausible, but not sufficiently grounded in the empirical evidence collected in this study. For example, the proposal to develop “decision trees mapping project characteristics to alignment pathways” is a reasonable design suggestion, yet the paper does not present empirical patterns showing that practitioners actually reason through alignment choices in a way that such a structure would meaningfully support. Rather, this appears to be an author extrapolation from general survey responses. Similarly, the recommendation around “two-pass generation” is based on a suggestion attributed to a single practitioner, but is elevated into a broader research direction without evidence that this need or strategy generalizes across the participant pool. More broadly, the discussion sometimes shifts from reporting what respondents said to prescribing what future tools and workflows should look like, without a sufficiently strong evidentiary bridge between the two.
- If some respondents reported that the models or alignment pathways were not applicable to them (Q7), it raises questions about what qualified them to answer alignment-focused questions in the first place. For Q8, it is unclear why responses from participants who do not use PTA/FTA in practice were included in the analysis. Without direct experience with these pathways, the basis for judging their importance is questionable, and the paper should justify why such responses were not excluded.
- The paper claims that including open-ended questions (Q6, Q14) mitigates the internal validity threat of self-reported data not reflecting actual behavior. This reasoning is fundamentally flawed. Open-ended questions are themselves self-reported data collected from the same respondents, within the same instrument, at the same sitting.
- The paper needs a clearer background about functional and non-functional properties for better understanding.

**Summary:**

This paper presents a survey of 30 practitioners on how they prioritize functional versus non-functional properties when aligning LLMs for code generation, what factors influence pathway choices between PTA and FTA, and whether they want systematic guidance. They found that 75.9% of respondents view non-functional properties as equally important or context-dependent, that task requirements and compute resources dominate decision-making, and that all applicable respondents want guidance on pathway selection and balancing requirements. While the topic is timely and potentially valuable, the study in its current form feels preliminary with limited methodological rigor.

---

> ### Author Response · Authors · 2026-03-20
>
> We thank the reviewer for their constructive feedback.
>
> **Central Findings**
>
> We acknowledge that the individual findings may align with intuitions from traditional software engineering. However, our contribution lies in providing the first empirical validation of these expectations specifically in the context of LLM alignment for code generation. While it may seem predictable that practitioners care about non-functional properties, the alignment research community has predominantly optimized for functional correctness. Our study quantifies the gap between what practitioners actually need and what alignment research currently prioritizes, providing empirical validation for a shift in research direction.
>
> **Thematic Analysis Details**
>
> We would like to clarify that the paper does not claim thematic saturation. We describe our qualitative analysis as following Braun and Clarke's six-phase approach, which we applied to the open-ended responses (Q6, Q11, Q14). However, we agree that Section 3.4 could provide more procedural detail on how the thematic coding was conducted. In the revised manuscript, we will expand this section to describe the coding process more explicitly.
>
> **The 100\% Guidance Demand and Response Scale Design**
>
> We would like to clarify that the actual survey does include four response options for Q12 and Q13: "Very useful," "Somewhat useful," "Not very useful," and "Not at all useful," in addition to "Not applicable to my work." Table 1 in the manuscript unfortunately only listed the first two options, which we recognize created a misleading impression. We will correct Table 1 in the revised manuscript to reflect the full set of response options.
>
> **Non-Significant Results Interpreted as Evidence of Consistency**
>
> We accept this critique. A non-significant result in a small-$N$ study reflects insufficient statistical power to detect a difference, not evidence that no difference exists. We will revise the language throughout the paper to avoid framing non-significant results as positive evidence of consistency, and instead describe them as showing no statistically detectable difference given our sample size.
>
> **Snowball Sampling and Selection Bias**
>
> We acknowledge that our snowball sampling strategy introduces selection bias toward research-adjacent and technically specialized individuals. We will strengthen the discussion of this limitation in Section 3.5, explicitly noting how the recruitment strategy may have shaped responses and narrowed the applicability of our conclusions.
>
> **Confounded Trade-Off Scenario in Q5**
>
> Q5 was intentionally designed to reflect the kind of coupled trade-off practitioners face in practice, where improving non-functional properties typically comes at some cost to absolute performance. Isolating these variables would make the scenario less realistic. Moreover, Q5 is not the sole basis for our RQ1 findings. It is triangulated with Q4 (explicit priority ratings) and Q6 (qualitative domain-driven scenarios), both of which independently support the same pattern. We will clarify this triangulation more explicitly in the revised manuscript.
>
> **Discussion Recommendations Not Sufficiently Grounded in Evidence**
>
> We agree that some recommendations (e.g., the decision tree, two pass generation) extend beyond what the data directly supports. We will more clearly distinguish empirically supported findings from author proposed directions motivated by, but not validated by, the survey responses.
>
> **Inclusion of Respondents Who Reported N/A for Alignment Questions**
>
> N/A responses were already excluded where applicable. The 100\% guidance demand is reported among applicable respondents only ($N=25$ for Q12, $N=27$ for Q13), and starting point preferences for $N=24$. Participants selecting N/A for Q7 may not currently work on alignment but can still hold informed views, given that 23.3\% are familiar with alignment concepts and 46.7\% have hands on experience. We will make the filtering logic more clear in the revised manuscript.
>
> **Open-Ended Questions as Mitigation for Self-Report Bias**
>
> We would like to clarify that we do not claim open-ended questions independently validate the quantitative findings. Our statement is that they "provide qualitative context validating the quantitative patterns," meaning convergent evidence through a different response modality. Generating a free-text scenario requires different cognitive effort than selecting a pre-defined option, which is a standard form of within-method triangulation in survey research.
>
> **Clearer Background on Functional and Non-Functional Properties**
>
> We agree. In the revised manuscript, we will expand Section 2 with clearer definitions and concrete examples of both property types to ensure accessibility for all readers.